# Setup of an Extraction Method for the Analysis of Carotenoids in Microgreens

**DOI:** 10.3390/foods9040459

**Published:** 2020-04-08

**Authors:** Vito Michele Paradiso, Maria Castellino, Massimiliano Renna, Pietro Santamaria, Francesco Caponio

**Affiliations:** 1Department of Soil, Plant and Food Science, University of Bari Aldo Moro, Via Amendola 165/A, 70126 Bari, Italy; maria.castellino@uniba.it (M.C.); francesco.caponio@uniba.it (F.C.); 2Institute of Sciences of Food Production, National Research Council of Italy, Via Amendola 122/O, 70126 Bari, Italy; massimiliano.renna@ispa.cnr.it; 3Department of Agricultural and Environmental Science, University of Bari Aldo Moro, Via Amendola 165/A, 70126 Bari, Italy; pietro.santamaria@uniba.it

**Keywords:** microgreens, carotenoids, bioactive compounds, antioxidants, extraction, lettuce, linen

## Abstract

Microgreens are gaining increasing interest as a potential functional food due to their relevant contents of micronutrients and bioactive compounds, including carotenoids. Nevertheless, the analysis of carotenoids is inherently difficult, due to their thermal and chemical susceptibility, as well as to their varying polarity. From this point of view, extraction is the most critical step, compared to chromatographic separation and detection. Thus, the reliability of data on carotenoids should be guaranteed by a constant focus on analytical issues, with appropriate adaptations to each sample matrix. In this research, a specific extraction procedure for the analysis of carotenoids in microgreens was developed. Solvent composition, extraction time, solvent/sample ratio, and repeated extractions were evaluated. The obtained protocol showed recovery of 97.2%, limits of quantitation of 5.2 μg·g^−1^ for lutein and 15.9 μg·g^−1^ for β-carotene, as well as intra-day mean repeatability of 5.7% and inter-day mean repeatability of 4.7%.

## 1. Introduction

Traditionally used for garnishing gourmet dishes, microgreens have been reconsidered over the last years as basic ingredients in several types of dishes [1], as well as for their potential in enhancing human diets due to relevant contents in micronutrients and phytochemicals [2,3,4,5]. In this regard, several reviews have been published in the last years [6,7,8,9].

Carotenoids are among the phytochemicals present in microgreens in considerable amounts that can be significantly affected by various endogenous and exogenous factors [10]. Carotenoids are one of the major classes of phytochemicals, and their importance in diet is not only related to their role as vitamin A precursors, but also to their antioxidant anti-tumor activities and their role in gene function regulation, gap-junction communication, and hormone and immune modulation [11,12]. Moreover, they cannot be synthesized by animals and need to be consumed through diet [11]. In this context, vegetable sources of carotenoids are obtaining a great interest [13,14]. As a matter of fact, several papers have evaluated the carotenoid contents in microgreens, reporting results varying in a wide range. To this end, irrespective of absolute concentrations, green leaves show quite a constant qualitative carotenoid pattern, referred to as a chloroplast carotenoid pattern, with lutein (about 45%), β-carotene (25–30%), violaxanthin (10%), and neoxanthin (10%) as the most represented carotenoids. Lactucaxanthin is another major carotenoid in lettuce [13,15,16]. In particular, lutein, the most represented xanthophyll, has been determined in microgreens of different genotypes and grown under different conditions in amounts ranging from 13 to 191 mg·kg^−1^ on fresh weight [10,17,18,19,20,21]. In most cases, these contents are quite higher than those observed in common fruits and vegetables [22]. Exceptionally higher amounts (from 105.7 to 503.5 mg·kg^−1^, with a mean content of 291.6 mg·kg^−1^ of lutein) were reported by Brazaitytė et al. [23] in three *Brassicaceae* microgreens grown under different lighting conditions. Regarding β-carotene, the most abundant carotene, the ranges observed in literature are even wider: from 0.11 to 121 mg·kg^−1^ [10,17,18,19,20,21,23]. Also in this case, outstanding results have been reported in another study, in which contents up to 8592.2 mg·kg^−1^ on a dry weight basis (corresponding to 451.9 mg·kg^−1^ on fresh weight) were reported for several species of microgreens grown under controlled conditions [24]. The wide range of carotenoids content in vegetables can be explained by genetic variability (intra- and inter-species biodiversity), as well as by different growing conditions. Nevertheless, analytical issues, particularly in carotenoid extraction, should not be disregarded [25,26]. In fact, carotenoids are easily degradable by diverse factors, and show varying affinity towards extraction solvents, due to their wide range of polarity [27]. As an example, xanthophylls, being oxygenated molecules, can be extracted with polar solvents such as alcohols, acetone, and acetone/water mixtures, while carotenes are more easily extracted by non-polar solvents [25]. Therefore, a possible underestimation of certain carotenoids could have occurred depending on the extraction solvent adopted, such as in some studies wherein 80% aqueous acetone was used as an extraction solvent. Moreover, some undervalued phenomena (degradation, isomerization) could also occur when the extraction procedure involves overnight extractions or some analytical steps, such as saponification, intended for different carotenoid patterns (e.g., carotenoid esters of fruit) or detection methods (e.g., direct spectrophotometry without chromatographic separation) [26,28].

With all the above remarks as a starting point, the aim of the present study was to set up an optimized extraction of carotenoids focused on microgreens as a specific food matrix. The general goal was to critically evaluate the effects of solvent polarity, extraction time, solvent/sample ratio, and repeated extractions.

## 2. Materials and Methods

### 2.1. Materials and Reagents

Microgreens of *Lactuca sativa* L. Group *crispa* (cultivar ‘Bionda da taglio’) and *Linum usitatissimum* L. were used. Seeds were purchased from Riccardo Larosa Company (Andria, Italy). The selected species was one of those characterized in our previous papers [2,3], showing intermediate levels of carotenoids compared to other genotypes.

Acetone (>99.5%), ethanol (96%), methyl *tert*-butyl ether (MTBE) for HPLC (≥99.8%), ammonium acetate, β-Carotene (≥93%), and trans-β-apo-8′-carotenal (≥96%) were purchased from Sigma-Aldrich, Milan, Italy. Methanol for HPLC (99.9%) was purchased from Honeywell (Monza, Italy). Lutein was provided by Extrasynthese (Genay Cedex, France). Butylated hydroxytoluene (BHT) was purchased from Fluka (Honeywell, Bucharest, Romania).

### 2.2. Microgreens Production and Storage

Microgreens were grown according to Paradiso et al. [2]. Batch samples were obtained by pooling microgreens harvested from at least three growing trays, lyophilized and stored at −20 °C until analyzed.

### 2.3. Sample Pre-Treatments

Lyophilization was chosen as the best dehydration method for both storage and sample pre-treatment, since it does not cause thermal degradation of carotenoids [25]. No other physical pre-treatment was used to facilitate the release of carotenoids, since microgreens are characterized by tender tissues, with very a low fiber content [2].

### 2.4. Protection against Degradation

BHT (0.1%) was added to the extraction solvent to prevent carotenoid oxidation. Extraction was carried out in dim light and the extraction vessels were covered with aluminum foil in order to protect carotenoids from photodegradation and isomerization during extraction [25,27,29].

### 2.5. Optimization of the Carotenoids’ Extraction

Lyophilized samples were weighted (0.05 g) in test tubes covered with aluminum foil and added with the extraction solvent and *trans*-β-apo-8′-carotenal (40 mL, 1 g·mL^−1^) as internal standard. Cold acetone, either pure or in mixture with water, was used as extraction solvent [29]. After centrifugation (3000 *g,* 5 min) the acetone layer was collected, and the pellet was submitted to further extraction where provided. The extraction procedure was set up through the subsequent steps:i.Evaluation of the solvent polarity mixing acetone with varying amounts of water (acetone 70%, 80%, 90%, 100%);ii.Evaluation of different extraction times (30 s, 10 min, 1, 5, 24 h);iii.Evaluation of the solvent/sample ratio (4, 5, 6, 12 mL of solvent per 0.05 g of sample);iv.Evaluation of repeated extractions.

Saponification is another critical step during carotenoid extraction. This procedure is aimed to remove chlorophylls, in case they could interfere during HPLC separation, and to improve extraction and separation of esterified xanthophylls [25]. Yet, saponification has important side effects (carotenoid degradation and loss, isomerization and formation of artefacts, especially involving more polar carotenoids such as lutein, violaxanthin, and neoxanthin [30]), and is often considered unnecessary in leafy vegetables, in which carotenoids are not esterified [16], a fortiori when HPLC analysis obtains appropriate separation of chlorophylls [13,16,25]. Therefore, due to the abundance of evidence in the literature, saponification was avoided.

### 2.6. HPLC Analysis of Carotenoids

The extracts were filtered using a 0,45 µm nylon filter and immediately analyzed by HPLC-DAD (Agilent Technologies, 1260 Infinity, USA), in accordance with the procedures reported by Rasmussen et al. [31]. Chromatography was carried out on a C_30_ column (3 µm, 150 × 4.6 mm, YMC, Japan). The mobile phase consisted of two components: eluent A, methanol:MTBE:water (95:3:2, by volume, with 1.5% ammonium acetate in water) and eluent B, methanol:MTBE:water (8:90:2, by volume, with 1.0% ammonium acetate in water). The flow rate was 0.4 mL·min^−1^, the injection volume was 25 µL, and all carotenoids were monitored at 445 nm. The gradient procedure (10 °C), was as follows: Start at 100% solvent A; a 22 min linear gradient to 45% solvent A and 55% solvent B; an 11 min linear gradient to 5% solvent A and 95% solvent B; a 4 min hold at 5% solvent A and 95% solvent B; a 2 min linear gradient back to 100% solvent A; a 28 min hold at 100% solvent A.

Carotenoid identification was carried out by means of analytical standards (β-carotene and lutein), comparison with retention times in literature, and UV spectra examination. Carotenoid quantification was performed using calibration curves of lutein for xanthophylls (in the range 0.1–10 μg·mL^−1^), β-carotene for carotenes (in the range 0.5–10 μg·mL^−1^), trans-β-apo-8′-carotenal (in the range 0.5–6 μg·mL^−1^) for the recovery evaluation. The linearity of calibration curves, expressed as adjusted R^2^, was 0.999.

### 2.7. Method Validity

Recovery of the optimized method was evaluated according to the following formula [32], applied to the internal standard:R’_A_ = Q_A_(*yield*)/Q_A_(*orig*)
where QA(*orig*) is the known original and QA(*yield*) is the recovered quantity of the analyte A.

Intra-day repeatability was evaluated repeating a series of six extractions in the same day, while inter-day repeatability was evaluated repeating a series of three extractions in three consecutive days.

Limit of detection (LOD) and limit of quantitation (LOQ) were calculated on the basis of the signal-to-noise (S/N) ratio in HPLC analysis, with LOD = 3 × S/N and LOQ = 10 × S/N, and were reported as μg·g^−1^ of sample (dry weight), considering the percent recovery [33]. An experimental limit of quantitation (ELOQ), the minimum quantified amount extracted by an exhaust real sample matrix, was also reported.

The method validity was also checked by comparison with another method applied in literature for microgreens by Kyriacou et al. [24].

### 2.8. Statistical Analysis

All the extractions were carried out at least in duplicate. One-way (to evaluate the effect of solvent/sample ratio and repeated extractions) and two-way (to evaluate the effect of solvent and extraction time) analysis of variance (ANOVA), followed by honestly significant difference (HSD) Tukey’s test for multiple comparisons, were carried out using Origin Pro 2019 (OriginLab, Northampton, Massachusetts, USA).

## 3. Results and Discussion

### 3.1. Optimization of the Carotenoids’ Extraction

#### 3.1.1. Effect of Solvent Polarity and Extraction Time

Considering the differences in polarity of the carotenoids existing in foods and their consequent differing affinity towards polar and non-polar solvents [29], the choice of the extraction solvent should consider the type of food matrix and its typical carotenoid pattern. Acetone and hexane are the most commonly used solvents for carotenoid extraction from food matrices [25,29]. Literature regarding carotenoids in vegetables and microgreens mainly reports methods using either acetone mixed with water [10,19,23,34], or hexane [24] and hexane/toluene [18] combined with saponification. Regarding 80% acetone, its high polarity should be taken under examination for the possible underestimation of non-polar carotenes, as pointed out in the introduction. On the contrary, hexane is mostly indicated for the extraction of non-polar carotenes and esterified xanthophylls [25]. Yet, green leafy vegetables, including microgreens, present the typical chloroplast carotenoid pattern [16], for which polar solvents are generally used. Therefore, we chose acetone for these reasons, as well as for its tunable polarity by mixing with water, and pure cold acetone or acetone mixed with varying amounts of water (10–30%) were evaluated. Different extraction times were also evaluated, ranging from 30 s to 24 h, considering that some extraction protocols provide overnight contact [22,29].

The results are reported in Figure 1, while the results of Tukey’s test for multiple comparisons are reported in the Appendix A. Regarding polar xanthophylls (a–e), the use of acetone:water mixtures provided high extraction yields in one hour of sample –solvent contact. The response to solvent polarity was, as expected, related to the xanthophyll polarity: Xanthophylls with epoxide moieties (violaxanthin, neoxanthin, luteoxanthin) were extracted in higher amounts by 70% acetone. Lutein and lactucaxanthin, having diol structure, gave similar results with 70% and 80% acetone. Extraction times longer than 1 h with acetone:water mixtures caused xanthophyll losses, probably due to oxidative enzymes activation and to isomerization phenomena, such as epoxide-furanoid rearrangement (isomerization of 5,6-epoxy- to 5,8-epoxycarotenoids), as pointed out by the marked decrease of violaxanthin and corresponding increase of luteoxanthin, its corresponding furanoid [25,27,28]. Acetone with 10% water showed an intermediate behavior between acetone:water mixtures and pure acetone. Extraction carried out with pure acetone gave the highest extracted amounts of xanthophylls after 24 h of contact. The extracted amounts were in almost all cases higher than those obtained with the acetone:water mixture, with the exception of luteoxanthin, which was presumably formed de novo, as stated above, due to epoxide rearrangement. Regarding carotenes (f–h), the performances of the solvent were quite different. Due to their non-polar nature, carotenes were poorly extracted by acetone 70% and 80% mixtures. On the contrary, relevant amounts were extracted with acetone 90% and, above all, with pure acetone. The highest extracted amounts were obtained after 24 h extraction. These results suggest the possibility that some low β-carotene contents reported in literature for microgreens could derive from an underestimation of carotenes due to the polarity of the adopted solvent (acetone 80%).

#### 3.1.2. Effect of Solvent/Sample Ratio

The effect of solvent/sample ratio was also evaluated. Different volumes of cold acetone (4, 5, 6, and 12 mL) were tested on the same amount of sample (0.05 g).

Figure 2 reports the results for three carotenoids (violaxanthin, lutein, and β-carotene), the most abundant and characterized by decreasing polarity. For all three molecules, the extracted amounts after 24 h extraction were comparable but with significant differences. The highest amounts were extracted using 5 mL of solvent for 0.05 g of sample. Higher volumes of solvent determined poorer extraction. This could be due to the fact that, during agitation, mechanical friction could have facilitated the extraction of carotenoids, while this effect would have been reduced by using higher volumes of solvent.

#### 3.1.3. Repeated Extractions

Since a single extraction step did not provide sufficient recovery, as pointed out by recovery data on the internal standard (data not shown), series of repeated extraction were evaluated. Repeated extraction is very common in carotenoid analysis [29]. This allowed to opt for shorter extraction times and to avoid possible analyte degradation during long-term analysis, observed by simulated extractions with a standard solution of β-carotene instead of microgreen sample (with a degradation higher than 18%; data not shown). For the repeated extractions testing, extracts of each step were injected separately to evaluate the contribution of each step to overall extraction.

Extraction steps of one hour were carried out, considering that the extraction time (Figure 1) could extract large amounts of carotenes and significant amounts of xanthophylls. Two extraction steps with 5 mL of cold acetone, followed by a step with 5 mL of acetone 70% to extract residue xanthophylls were tested (Figure 3). The last step was effective for polar carotenoids. A slight improvement of extracted amounts was observed after a slight hydration of the sample before analysis [29], while longer extraction times caused a decrease in extracted amounts.

Extraction times higher than 2 h per step were not considered in order to keep reasonable analysis duration. Therefore, the volumes of extracting solvents (10 mL of acetone and 5 mL of acetone 70%) were split into four aliquots (4, 3, and 3 mL of acetone and 5 mL of acetone 70%) and four 1-hour extraction steps were performed.

The results are reported in Figure 4, in comparison with those obtained with the same amount of solvents divided into three extraction steps. This change determined relevant increases in the extracted amounts of both carotenes and xanthophylls. Therefore, this procedure was adopted and considered for the evaluation of recovery and repeatability. Further extractions with 5 mL aliquots of solvents (either a fourth pure acetone step, before acetone 70% step, or a fifth step with acetone 70%) provided limited increases in extracted carotenoids, in the ranges 0–1.5% and 0–3.7%, respectively. Though a third extraction could provide a more efficient procedure, we believe that environmental issues should not be disregarded even in analytical chemistry; therefore, this increase in solvent volumes could be avoided without relevant analyte losses [35,36]. Figure 5 reports the flowchart of the analytical protocol.

A typical chromatographic separation is reported in Figure 6.

#### 3.1.4. Method Validity

The recovery of the internal standard is reported in Figure 7. The adopted extraction procedure allowed to recover 97.2% of the internal standard. Further extraction steps allowed further recoveries of 1.1% (acetone 100%) and 0.5% (acetone 70%).

The figures of merit of method validation are reported in Table 1.

Limits of detection (LOD) and quantitation (LOQ), calculated on the basis of S/N ratio, were quite below 10 μg g^−1^ for lutein, while the indices slightly exceed this threshold for β-carotene. Experimental limit of quantitation determined as the lowest real measure obtained by submitting to extraction an exhausted sample matrix substantially complied. Violaxanthin, lutein, and β-carotene were considered to be the most abundant and showed decreasing polarity. The C.V.% of repeatability tests ranged 4.1–5.3% (5.7% mean value) for inter-day repeatability and 4.4–6.9% (4.7% mean value) for inter-day repeatability and was considered satisfactory.

As a last step, the method developed was compared with another method from literature applied to carotenoids [24], involving a saponification step followed by hexane extraction. Besides lettuce microgreens, characterized by very tender tissues, microgreens of linen (*Linum usitatissimum* L.), characterized by tough tissues, were also submitted to carotenoid extraction. The results are reported in Figure 8.

As can be observed, the method developed in the present study provided higher amounts of extracted carotenoids compared to the method which involves saponification and hexane extraction. The largest differences could be observed for the more polar xanthophylls, probably underestimated when extraction was carried out with hexane, compared to carotenes, for which the differences between the two methods were less marked. Moreover, differences between the methods were more pronounced for lettuce microgreens compared to linen microgreens. This is because saponification probably caused a certain degradation of carotenoids when applied to more tender tissues. Therefore, the method developed appeared to be suitable to be applied to microgreens.

In conclusion, an effective protocol for the extraction and analysis of carotenoids from microgreens was setup in the present research. The protocol was developed considering several variables (i.e., solvent polarity, extraction time, solvent/sample ratio, repeated extractions) and was optimized on this matrix according to its typical carotenoid pattern, characterized by a wide range of polarity, and possible degradation/isomerization phenomena that could occur. Good recovery, mean repeatability, and limits of detection and quantitation characterized this method, which proved to be more in the extraction of carotenoids from the delicate tissues of microgreens, even compared to another method from literature. The critical development of a reliable analytical method can allow for affordable nutritional data on such an emerging food, which is claiming increasing attention for its functional potential and for its suitability for tailored nutrition.

## Figures and Tables

**Figure 1 foods-09-00459-f001:**
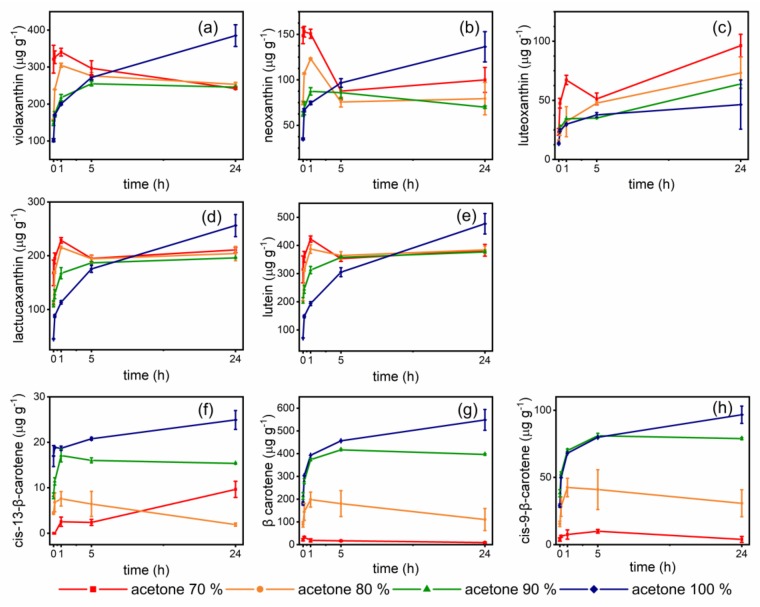
Carotenoids extracted from lettuce microgreens with different acetone:water mixtures and varying extraction times. (**a**–**e**), xanthophylls; (**f**–**h**), carotenes. Error bars indicate standard deviation. Results of ANOVA and post-hoc Tukey’s test are reported in Appendix A.

**Figure 2 foods-09-00459-f002:**
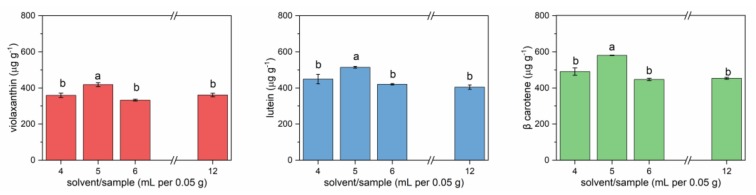
Carotenoids extracted from lettuce microgreens with different solvent/sample ratios. Error bars indicate standard deviation. Different letters mean a significant difference at *p* < 0.05.

**Figure 3 foods-09-00459-f003:**
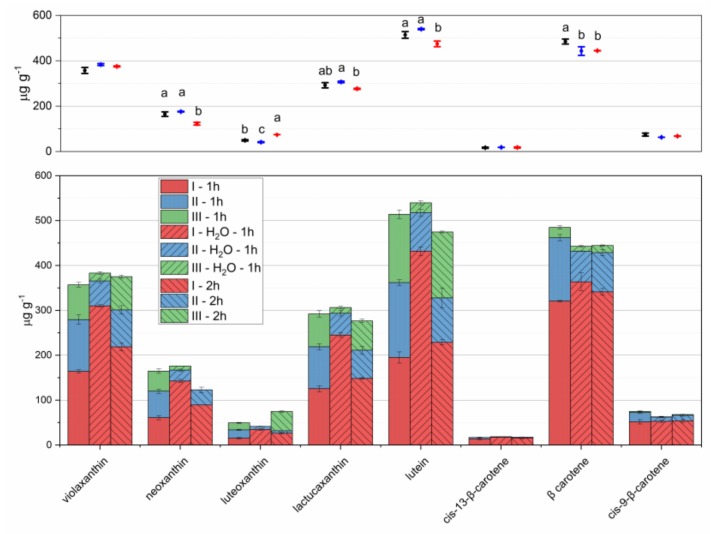
Carotenoids extracted from lettuce microgreens with three repeated extraction steps (I–III in stacked bars), with 1-hour extractions (left bars); initial sample rehydration and 1-hour extractions (central bars); 2-hour extractions (right bars). For steps I and II, 5 mL of pure acetone were used; for step III 5 mL of acetone 70% were used. Means and standard deviations of each step are reported in the stacked bar plot of lower panel; means and standard deviations of overall extraction are reported in the scatter plot of upper panel. Different letters mean a significant difference at *p* < 0.05.

**Figure 4 foods-09-00459-f004:**
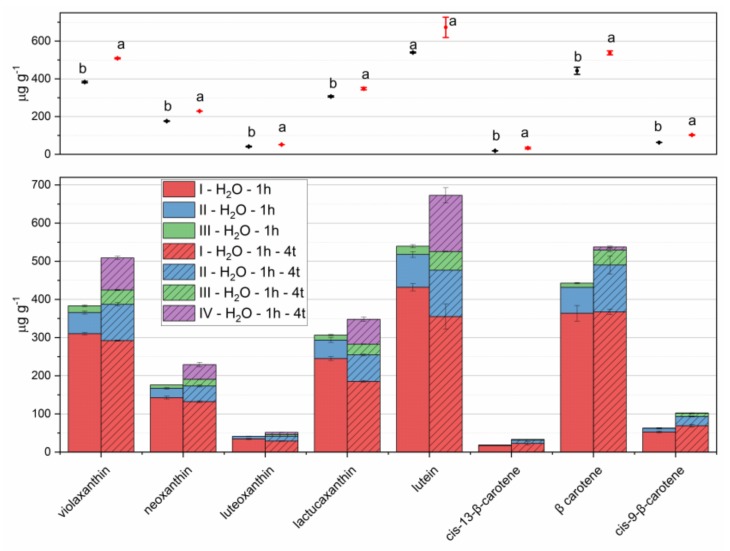
Carotenoids extracted from lettuce microgreens with three repeated 1-hour extraction steps (I–III in left stacked bars, steps I and II with 5 mL of pure acetone, step III with 5 mL of acetone 70%); with four repeated 1-hour extraction steps (I–IV in right stacked bars, step I with 4 mL of pure acetone, steps II and III with 3 mL of pure acetone, step III with 5 mL of acetone 70%). Means and standard deviations of each step are reported in the stacked bar plot of lower panel; means and standard deviations of overall extraction are reported in the scatter plot of upper panel. Different letters mean a significant difference at *p* < 0.05.

**Figure 5 foods-09-00459-f005:**
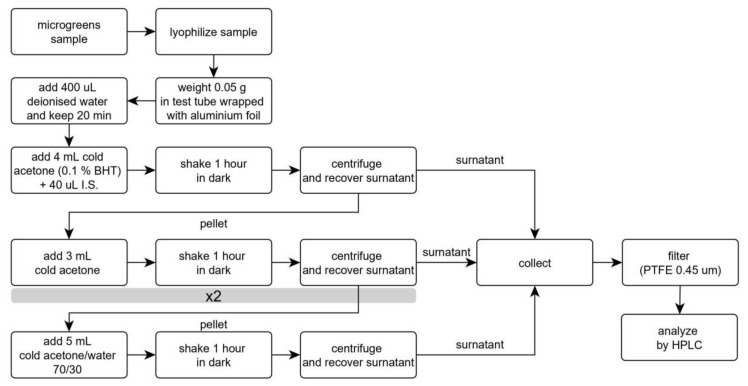
Flowchart of the extraction protocol for the analysis of carotenoids in microgreens.

**Figure 6 foods-09-00459-f006:**
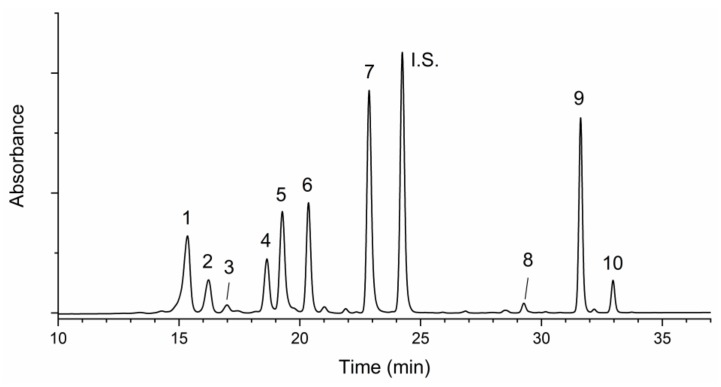
Typical chromatogram of carotenoids extracted from a microgreen sample of *Lactuca sativa* L. Group *crispa* (cultivar ‘Bionda da taglio’). For chromatographic and detection conditions, please see Section 2.8 in the text. 1, violaxanthin; 2, neoxanthin; 3, lutheoxanhin; 4, lactucaxanthin; 5, chlorophyll b; 6, lutein; 7, chlorophyll a; I.S., internal standard; 8, cis-13-β- carotene; 9, β-carotene; 10, cis-9-β-carotene.

**Figure 7 foods-09-00459-f007:**
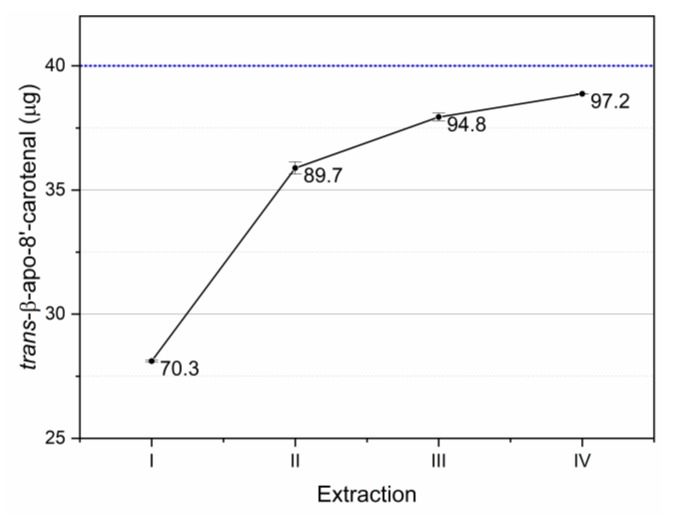
Recovery of the internal standard added to the sample before extractions. Blue reference line indicates the spiked amount. Data labels indicate the cumulative percent recovery after each extraction step of the tested extraction protocol. Error bars indicate standard deviation.

**Figure 8 foods-09-00459-f008:**
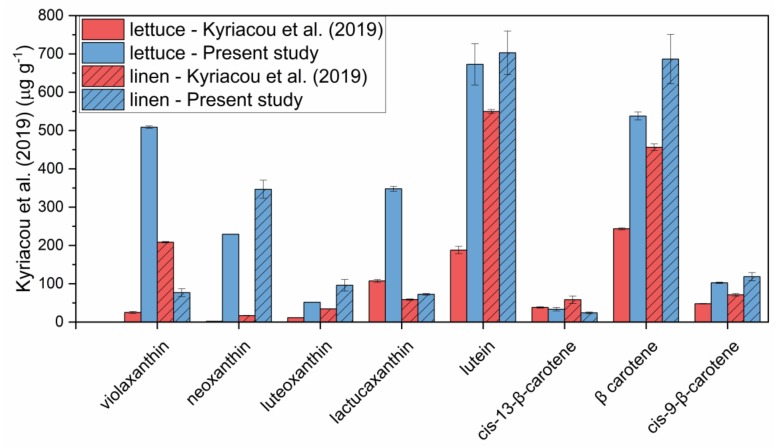
Carotenoids extracted from lettuce and linen microgreens with the method developed in the present study and with a method in literature [24]. Error bars indicate standard deviation.

**Table 1 foods-09-00459-t001:** Figure of merit of the optimized method (intra-day repeatability, n = 6; inter-day repeatability n = 3 × 3; limits are expressed as μg g^−1^ of sample on dry weight basis).

	Internal Standard	Violaxanthin	Lutein	β-Carotene
Recovery	97.2%			
Linearity (Adjusted R^2^)	0.999		0.999	0.999
Limit of detection (LOD)			1.6 μg g^−1^	11.3 μg g^−1^
Limit of quantitation (LOQ)			5.2 μg g^−1^	15.9 μg g^−1^
Experimental limit of quantitation (ELOQ)			8.4 ± 0.5 μg g^−1^	9.1 ± 1.1 μg g^−1^
Intra-day repeatability (C.V.%)		4.4%	5.7%	6.9%
Inter-day repeatability (C.V.%)		4.1%	4.8%	5.3%

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
