# Peer review of "Setup of an Extraction Method for the Analysis of Carotenoids in Microgreens"

_foods, 2020, doi:10.3390/foods9040459_

Round 1

Reviewer 1 Report

Dear Author Thank you for the submission. The paper is well written and organized. It is of interest and very detailed in methodology. Can you add the chromatic separation of carotenoids in the discussion and the validity of the method. Once this is done, please resubmit for review.

Author Response

Authors sincerely thank the referee for helping to improve this manuscript.

Please find below the detailed response to referee's suggestions.

Dear Author Thank you for the submission. The paper is well written and organized. It is of interest and very detailed in methodology. Can you add the chromatic separation of carotenoids in the discussion and the validity of the method. Once this is done, please resubmit for review.

  • A typical chromatogram was included in the manuscript as Figure 6. Method validity was assessed by figures of merit (Table 1) and comparison with another method from literature (Figure 8). Please see text in lines 268-304

Best regards

Vito M. Paradiso, on behalf of all Authors

Reviewer 2 Report

The manuscript entitled “Setup of an extraction method for the analysis of carotenoids in microgreens” by Paradiso and co-workers describes the optimization of the extraction procedure of carotenoids present in microgreen for their further analysis. The presented method is characterized by good recovery and repeatability as described by the authors. In my opinion, the manuscript should be better presented to be accepted in Foods. There are some drawbacks, the significance of content is not precisely determined, results are not clearly presented, conclusions are too short. Presented manuscripts may be accepted after major revision. My comments are presented below. Please provide the explanation for all of them, make changes in the text. Correct language. Major concerns: - Abstract – abstract should better presents the main goal of the work. The significance of the presented topic should be exposed. - Abstract – The obtained protocol showed recovery of 97.2%, intra-day mean repeatability of 5.7% and inter-day mean repeatability of 4.7% - the 5.7 value is presented in Table 1 (page 8), however, inter-day mean repeatability of 4.7% is not presented anywhere. What is this value? Introduction – introduction should shows the importance of the carotenoids determination in food, their role and significance. Introduction, page 1, lines 30-31 - (Samuolienė et al., 2016). – it should be cited as [10]. The final part of Introduction should inform about the applicability of the performed investigation. Materials and methods – please explain why the Lactuca sativa was used as a model microgreen. Materials and methods, section 2.3 – please explain why aluminium foil is important. Materials and methods, section 2.7 – solvent A, solvent B – in my opinion, eluent is much more precise. Materials and methods, section 2.7 – Chromatography was carried out on a C30 column (3 µL, 150 mm x 4.6 mm, YMC, Japan – what parameter describes 3 µL? Materials and methods, section 2.7 – Carotenoid quantification was performed using calibration curves of lutein (for xanthophylls), β-carotene (for carotenes), trans-β-apo-8’-carotenal for the recovery evaluation – what concentration did you use to obtain the calibration curves? b-carotene – should be β-carotene (page 3, line 117) Results and discussion – “Acetone and hexane are the most commonly used solvents for carotenoid extraction from food matrices [22,25]” “Our choice fell on acetone, and pure cold acetone or acetone mixed with varying amounts of water (10-30%) were evaluated” Briefly describe previously proposed methods. What is the advantage of the proposed method? What are the differences between proposed method and previously described using acetone? Page 5, line 174 – should be β-carotene Figure 3 caption - See text for solvents and volumes of each step – in my opinion it should be pointed here, in figure caption. It may facilitate graph analysis. There are two Figures 5 (page 7). Please correct. Conclusions in my opinion are too short, do not fit with the results. The importance of presented topic is not fully exposed. Correct the style of references, check the font style, page numbers (see reference 15).

Author Response

Authors sincerely thank the referee for helping to improve this manuscript.

Please find below the detailed response to referee's suggestions.

The manuscript entitled “Setup of an extraction method for the analysis of carotenoids in microgreens” by Paradiso and co-workers describes the optimization of the extraction procedure of carotenoids present in microgreen for their further analysis. The presented method is characterized by good recovery and repeatability as described by the authors. In my opinion, the manuscript should be better presented to be accepted in Foods. There are some drawbacks, the significance of content is not precisely determined, results are not clearly presented, conclusions are too short. Presented manuscripts may be accepted after major revision. My comments are presented below. Please provide the explanation for all of them, make changes in the text.

Correct language.

  • Language was revised

Major concerns:

- Abstract – abstract should better presents the main goal of the work. The significance of the presented topic should be exposed.

  • Abstract was revised improving the description of the main goal (lines 14-24)

- Abstract – The obtained protocol showed recovery of 97.2%, intra-day mean repeatability of 5.7% and inter-day mean repeatability of 4.7% - the 5.7 value is presented in Table 1 (page 8), however, inter-day mean repeatability of 4.7% is not presented anywhere. What is this value?

  • Please consider that 5.7% and 4.7% were the mean values of intra-day and inter-day repeatability, respectively, of the three carotenoids reported in Table 1. Mean values are now reported in the text (lines 282-283) for better clarity

Introduction – introduction should shows the importance of the carotenoids determination in food, their role and significance.

  • Authors thank referee for the suggestion. A paragraph on the role and importance of carotenoids in foods has been added (lines 33-39)

Introduction, page 1, lines 30-31 - (Samuolienė et al., 2016). – it should be cited as [10].

  • We thank the referee for the attention paid to the manuscript. The reference was updated

The final part of Introduction should inform about the applicability of the performed investigation.

  • The final part of the Introduction was rewritten (lines 63-71)

Materials and methods – please explain why the Lactuca sativa was used as a model microgreen.

  • The selected species was one of those characterized in our previous papers, showing intermediate levels of carotenoids compared to other genotypes. This explanation was included in the text (lines 76-77)

Materials and methods, section 2.3 – please explain why aluminium foil is important.

  • Aluminium foil was used in order to protect carotenoids from photodegradation and isomerization during extraction. The explanation was included in the text (lines 93-95)

Materials and methods, section 2.7 – solvent A, solvent B – in my opinion, eluent is much more precise.

  • We thank the referee. The terms were corrected as suggested

Materials and methods, section 2.7 – Chromatography was carried out on a C30 column (3 µL, 150 mm x 4.6 mm, YMC, Japan – what parameter describes 3 µL?

  • We apologize for the misprint and thank the referee for the attention paid to the manuscript. The parameter described is the particle size of the column, and the measurement unit are micrometres (mm). The mistake was corrected

Materials and methods, section 2.7 – Carotenoid quantification was performed using calibration curves of lutein (for xanthophylls), β-carotene (for carotenes), trans-β-apo-8’-carotenal for the recovery evaluation – what concentration did you use to obtain the calibration curves?

  • We thank the referee. The data requested were reported (lines 127-131)

b-carotene – should be β-carotene (page 3, line 117)

  • We thank the referee. The typo was amended

Results and discussion – “Acetone and hexane are the most commonly used solvents for carotenoid extraction from food matrices [22,25]” “Our choice fell on acetone, and pure cold acetone or acetone mixed with varying amounts of water (10-30%) were evaluated” Briefly describe previously proposed methods. What is the advantage of the proposed method? What are the differences between proposed method and previously described using acetone?

  • The paragraph was rewritten according to the referee’s suggestion (lines 157-171)

Page 5, line 174 – should be β-carotene

  • The typo was amended

Figure 3 caption - See text for solvents and volumes of each step – in my opinion it should be pointed here, in figure caption. It may facilitate graph analysis.

  • Solvents and volumes were reported in the captions of Figures 3 and 4, as requested.

There are two Figures 5 (page 7). Please correct.

  • We apologize for the negligence. The last Figure was correctly numbered as 7.

Conclusions in my opinion are too short, do not fit with the results. The importance of presented topic is not fully exposed.

  • Conclusions were rewritten (lines 305-314).

Correct the style of references, check the font style, page numbers (see reference 15).

  • We checked references. We now believe they correspond to the style of Foods

Round 2

Reviewer 2 Report

In my opinion the manuscript in presented form may be accept for publication in Foods. Authors presented the responses for all of my comments.